# LESS-PHARMA Study: Identifying and Deprescribing Potentially Inappropriate Medication in the Elderly Population with Excessive Polypharmacy in Primary Care

**DOI:** 10.3390/ijerph192013241

**Published:** 2022-10-14

**Authors:** Xisco Reus, Maria Lluisa Sastre, Alfonso Leiva, Belén Sánchez, Cristina García-Serra, Ignatios Ioakeim-Skoufa, Caterina Vicens

**Affiliations:** 1Balearic Health Service IbSalut Son Serra-La Vileta Healthcare Centre, 07013 Palma, Spain; 2Research Network on Chronicity, Primary Care, and Health Promotion (RICAPPS), Balearic Islands Health Research Institute (IdISBa), 07120 Palma, Spain; 3Balearic Health Service IbSalut, Reseach Unit Primary Care Mallorca, 07013 Palma, Spain; 4Department of Drug Statistics, Division of Health Data and Digitalisation, Norwegian Institute of Public Health, 0213 Oslo, Norway; 5Drug Utilization Work Group, Spanish Society of Family and Community Medicine (semFYC), 08009 Barcelona, Spain; 6EpiChron Research Group, IIS Aragón, Miguel Servet University Hospital, 50009 Zaragoza, Spain

**Keywords:** potentially inappropriate medication, polypharmacy, deprescribing, elderly

## Abstract

Potentially inappropriate medication (PIM) increases adverse drug reactions and mortality, especially in excessively polymedicated patients. General practitioners are often in charge of this process. Some tools have been created to support them in this matter. This study aimed to measure the amount of potentially inappropriate medication among excessively polymedicated patients using several supporting tools and assess the feasibility of these tools in primary care. Several explicit deprescribing criteria were used to identify potentially inappropriate medications. The level of agreement between all the criteria and the acceptance by the general practitioner (GP) was also measured. We analysed whether the drugs proposed for deprescribing were eventually withdrawn after twelve months. The total number of drugs prescribed was 2038. Six hundred and forty-nine drugs (31.8%) were considered potentially inappropriate by at least one of the tools. GPs agreed with the tools in 56.7% of the cases. In a 12-month period, 109 drugs, representing 29.6% of the drugs that GPs agreed to deprescribe, were withdrawn. Elderly excessively polymedicated patients accumulated a great number of PIMs. The use of deprescribing supporting tools, such as explicit criteria, is feasible in primary care, and these tools are well accepted by the GPs. However, eventual withdrawal was carried out in less than half of the cases.

## 1. Introduction

Polypharmacy and the concurrent use of potentially inappropriate medications have become more prevalent. Elderly patients and patients with multiple chronic conditions are at a higher risk of having multiple prescriptions [1,2]. Pharmacological treatment is one of the most common medical interventions in many diseases, along with surgical treatment and lifestyle modifications. Many new drugs have been developed over the past decades. The increase in life expectancy, the progressive ageing of the population, the potential for many diseases to become chronic, and the lack of clinical guidelines for stopping medications are the main reasons for the increase in polypharmacy.

Although there is no official definition of polypharmacy, most studies in this field have adopted the numeric definition of five or more drugs consumed over 90 days. Consumption of ten or more drugs is known as excessive polypharmacy [3].

According to the Spanish National Health Survey of 2012, the prevalence of polypharmacy was 36% among patients over the age of 65, reaching 47% among patients over the age of 85 [4]. During the past decade, the number of patients receiving five or more drugs increased threefold, and the number of patients with excessive polypharmacy increased 10-fold [5]. Similar results were found across Europe, where the prevalence of polypharmacy in older adults ranged from 26.3% to 39.9% [6].

Polypharmacy is related to lower adherence; an increase in hospital admissions; and an increase in adverse drug reactions (ADR), such as falls, bleeding, kidney failure, cognitive and functional impairment, and delirium [7,8]. It is estimated that 87% of elderly polymedicated patients (PPs) suffer from ADRs [9]. Polypharmacy is also related to an increase in mortality rates directly proportional to the number of drugs, reaching a relative risk (RR) of 2 in excessively Polymedicated patients. It is challenging to know how much of this increase is due to comorbidity and how much is due to polypharmacy itself [10].

Regarding costs, the cost of polypharmacy is derived not only directly from the drugs but also from the indirect costs derived from hospital admissions due to ADRs. In Spain, ADRs in patients with polypharmacy had an impact of EUR 14.5 million in 2003 [11], but the cost can be much higher, reaching USD 180 billion in the USA [12].

The term potentially inappropriate medication (PIM) refers to drugs that are no longer needed because of a lack of effectiveness, adverse reactions, a short life expectancy, etc. [13]. One in five prescriptions is estimated to be a PIM in older adults [14]. Among patients with polypharmacy, PIMs are found in more than half of patients [15].

The process of identifying and withdrawing drugs to reduce polypharmacy and improve health outcomes is known as deprescribing [16]. It might be logical to think that a reduction in polypharmacy could be translated to a decrease in mortality. However, despite a tendency to reduce mortality, a recent systematic review only found a statistically significant reduction in mortality when the research team performed patient-specific interventions and not when training programs when given to the physicians [17]. Although the impact of deprescribing on mortality is still unclear, evidence suggested that individualised interventions to reduce inappropriate polypharmacy are safe [17].

Medication review takes into account the drug regimen, clinical assessment criteria, and the patient’s goals and priorities. It is a complex process, as multiple factors interact to optimise medication at the levels of the healthcare system, healthcare professionals, and patients. Healthcare professionals need to acquire specific competencies to be able to carry out this procedure.

Medication reviews can be carried out in many settings, but they are usually performed by general practitioners (GPs). Thus, the identification of PIMs and the deprescribing process rely on them; these tasks are very time-consuming in scope with an increasingly high demand for services. Several tools have been created to help physicians with this process: online tools, such as MedStopper and CheckTheMeds [18,19], and explicit criteria, such as BEERS [20], PRISCUS [21], STOPP/START (screening tool of older people’s prescriptions/screening tool to alert to right treatment) [22], and LESS-CHRON [23]. Some studies have concluded that these tools effectively help physicians deprescribe [24,25].

This study aimed to identify the use of PIM in patients older than 75 years in a primary healthcare centre and to determine the feasibility of using some deprescribing support tools to help physicians review and deprescribe PIMs.

## 2. Materials and Methods

### 2.1. Study Design

This was a prospective longitudinal study conducted between September 2019 and November 2020 in primary care. The study was conducted in a primary healthcare centre (PHC) in Son Serra-La Vileta (Spain), covering a population of 26,930 inhabitants with 1677 aged 75 years and over.

### 2.2. Participants and Recruitment

Excessively polymedicated patients were identified in September 2019 from the RELE (e-prescription, Receta Electrónica in Spanish) database. The RELE is the e-prescription system and database used in the Balearic Islands health services (Ib-salut). The database includes information about all primary care prescriptions dispensed in community pharmacies, including the issuing date, pharmaceutical product (active ingredient and brand name), dose, and treatment duration. 

Excessive polypharmacy was defined as the intake of ten or more drugs marked as “chronic” on the electronic prescription sheet. Combined medications were counted as the number of drugs that were included. Topic, ophthalmic, and inhaled medications were included if they met the chronicity criteria. Prescriptions marked “on-demand” (described as pro re nata or taken only when needed) were excluded, and so were diapers, dressing materials, and nutritional complexes. Homoeopathic products and over-the-counter drugs were not registered. All excessively polymedicated patients 75 years or older were reviewed.

The exclusion criteria were regular follow-ups in the private sector and lacking a blood test in the past year.

### 2.3. Data Collection and Procedure

The selected patients were included in the Data Collection Logbook (DCL). The descriptive variables recorded were age, sex, weight, high blood pressure (reported as the average of the last three measurements in the clinical history), creatinine, total cholesterol, and low-density lipoprotein cholesterol (LDLc), with a list of all chronic comorbidities and drugs. Other variables required by the deprescribing tools were also registered, such as glycosylated haemoglobin (HbA1c) in diabetic patients, urate and last gout episode in hyperuricemic patients, diaper use in patients with urinary incontinence, and the Barthel index in patients with calcium and/or vitamin D supplement intake. Whether the patients with a prescription of benzodiazepines were consulted for insomnia in the past month or anxiety in the past six months was also recorded. For the patients with prescriptions for antidepressants, a consult for any mood disorder in the past six months was recorded. All these items were required by the deprescribing tools afterwards to apply the deprescribing criteria.

Drug utilisation data were introduced in CheckTheMeds (version 3.6.4 owned by CheckTheMeds Technology SL in Almería, Spain), an online deprescribing tool that processes all information related to drugs [19]. This tool applies the deprescribing criteria (BEERS, STOPP/START, and PRISCUS) automatically and gives information on overdosing, underdosing, duplicity, cardiovascular risk, and anticholinergic burden. It requires the introduction of patient data, blood test results, comorbidities, and medication to create a deprescribing proposal with all the PIMs identified. Since CheckTheMeds did not include the LESS-CHRON criteria, they were applied manually to each patient, adding the PIMs found to the prior proposal.

Once the deprescribing proposal with all the PIMs per patient was made, the research team members had a meeting with every GP of the PHC to discuss the deprescribing proposal of their patients’ quotas. The GPs were asked if they agreed with the deprescribing proposal for each PIM in each of their patients. The criteria that all GPs used to decide if they agreed with the proposal were based on a patient-centred, integrated care model that took into consideration the clinical profile, functional status, mental health, and socio-economic status, also considering patient preferences. In the case of an affirmative answer, the GP would compromise to try to deprescribe the drug within 12 months. In the case of a negative answer, the reason was registered.

### 2.4. Primary and Secondary Outcomes

The primary outcomes were the number of PIMs for each patient and therapeutic groups more often considered as PIMs, as well as the proportion of PIMs with GP agreement to be withdrawn and the proportion of PIMs deprescribed after 12 months of the researcher–GP meeting.

The secondary outcome was a description of the excessively polymedicated population aged 75 years and older in terms of age, sex, mass body index (MBI), comorbidities. and control of cardiovascular risk factors (hypertension, diabetes, and dyslipidaemia).

### 2.5. Statistical Analysis

The results of the 12-month assessments carried out between November 2019 and November 2020 are presented. All analyses were conducted using SPSS version 21.0. Continuous data are expressed as means ± SD, nonparametric variables are expressed as medians and interquartile ranges (IQRs), and categorical data are expressed as frequencies. We carried out the Pearson’s correlation to analyse the association between the number of drugs prescribed and the number of comorbidities; a *p*-value of 0.05 was considered statistically significant. 

## 3. Results

### 3.1. Participants

Of 1677 patients 75 years or older, 288 (17,2%) were excessively Polymedicated patients, and 168 (58.3%) were included in our study after applying the inclusion and exclusion criteria. Two patients were lost to follow-up at six months (one deceased and one was transferred to another PHC). The characteristics of the study population are described in Table 1. The average number of drugs per patient was 12.24 ± 2.17, and the average number of PIMs per patient was 3.86 ± 2.56. The median number of comorbidities was 8 (7–9). No differences in sex were found in these three variables. Hypertension, dyslipidemia, and diabetes were the most common comorbidities. No relationship between age and the number of drugs prescribed per patient was found. A positive, statistically significant association (*p* < 0.001) between the number of drugs prescribed and the number of comorbidities per patient was observed (Pearson’s correlation coefficient: 0.332).

Almost all excessively Polymedicated patients were taking an antihypertensive drug (164/168, 97.6%), 148/168 (88.1%) were taking a PPI, 130/168 (77.4%) were taking a lipid-lowering drug, and 125/168 (75%) were taking an antiplatelet/anticoagulant.

Table 2 describes the level of control of excessively Polymedicated patients for some pathologies. Starting with hypertensive patients, nearly 75% had blood pressure (BP) under 150/90, and 63.7% were on three or more antihypertensive drugs. Similarly, 75% of diabetic patients had HbA1C under 8%, and 58% were on two or more drugs. Patients diagnosed with dyslipidaemia had normal cholesterol values in 77.4% of the cases.

### 3.2. Potentially Inappropriate Medication

Table 3 shows the number of drugs prescribed per therapeutic group (TG) and the number of drugs proposed to be deprescribed by the deprescribing supporting tools.

Overall, 2038 drugs were prescribed. The TGs most commonly prescribed were antihypertensive drugs (22.9%), antidiabetic drugs (8.9%), and vitamins/supplements (7.5%).

Six hundred and forty-nine drugs (31.8%) were proposed to be deprescribed by a deprescribing tool. The tool that suggested the most drugs to be deprescribed was LESS-CHRON with 339 drugs (16.6%). The TGs most frequently proposed for deprescribing were hypnotics/anxiolytics (97.1%), antidepressants (83.7%), and urine incontinence drugs (72.7%).

We analysed the level of agreement between tools with Cohen’s kappa coefficient (*κ*). The tools with the highest level of agreement were BEERS with STOPP and BEERS with LESS-CHRON, showing a *κ* of 0.43 and 0.30 respectively.

### 3.3. Feasibility of the Tools

The GPs agreed to deprescribe the drugs proposed by the tools in 56.7% of cases. LESS-CHRON was the tool with the highest level of agreement (67%), followed by STOPP (56.7%), PRISCUS (50%), and BEERS (32.2%). The main reasons for disagreement with the deprescribing proposal were bad control of the pathology for which the drug was prescribed; the drug was prescribed or followed up by another specialist; or deprescribing was already tried in the past for the drug without success, which was especially observed with benzodiazepines and antidepressants

The TGs with the highest levels of agreement by the GPs to deprescribe were the lipid-lowering drugs, with agreement in 42 of the proposed cases (87.5%); urate-lowering drugs, with agreement in 16 cases (84.2%); and antiplatelets/anticoagulants, with agreement in 26 cases (76.5%) due to acetylsalicylic acid in primary prevention. Other TGs with a high level of agreement to be deprescribed were antidiabetic drugs, with agreement in 21 cases (63.6%); hypnotics/anxiolytics, with agreement in 41 cases (62.1%), vitamins/supplements, with agreement in 26 cases (47.3%); and PPIs, with agreement in 28 cases (46.7%).

Twelve months after the deprescribing proposal was discussed with the GPs, 109 drugs had been withdrawn, representing 16.8% of all prescribed PMI drugs and 29.6% of the drugs that GPs agreed to deprescribe. BEERS was the tool that accomplished the deprescription of the most drugs with a 36.8% success rate, followed by STOPP (34.8%), PRISCUS (30.8%), and LESS-CHRON (30%).

The most frequently deprescribed TGs were urate-lowering drugs, which were deprescribed in 42.1% of the proposed cases, followed by antidiabetic drugs (33.3%) and antipsychotic drugs (28.6%). Other TGs with a high level of deprescribing were antiplatelets/anticoagulants (26.5%), lipid-lowering drugs (25%), vitamins/supplements (21.8%), and antiarrhythmics (20%).

Figure 1 summarizes the percentage of prescribed medication, potentially inappropriate medication, the level of agreement of the GPs with the deprescribing proposal, and the drugs eventually deprescribed at 12 months per therapeutic group.

## 4. Discussion

One of the main findings of this study on identifying PIMs in excessively Polymedicated patients is that almost four drugs per patient were considered PIMs. The top three comorbidities were hypertension, dyslipidemia, and diabetes. Accordingly, among the most prescribed TGs, we found antihypertensive, antidiabetic, and lipid-lowering drugs. Excessively Polymedicated patients were found to have a high degree of control over cardiovascular risk factors. During the past decade, most medical guidelines have recommended a less strict degree of control in the elderly population due to a reduction in the benefits and an increased risk of adverse effects. Despite the recommendations, some difficulties are found in daily practice for these situations, such as the fear of worsening the patient’s condition or reluctance to withdraw a drug that they have been taking for decades. The risk of developing an ADR is exceptionally high in excessively Polymedicated patients; thus, a less aggressive approach for cardiovascular risk factors should be considered in these patients and always after individual assessment. Despite the fact that the actual recommendation for elderly patients is a systolic BP under 150 mmHg, more than half of patients had systolic BP levels under 140 mmHg, and almost two-thirds were on three or more antihypertensive drugs. It was also observed that more than three-quarters of patients with dyslipidemia were in a normal cholesterol range. However, half of them were taking lipid-lowering drugs for primary prevention.

Almost a third of all prescribed medications were considered PIMs by at least one of the tools, and GPs agreed with more than half of the proposed drugs. However, less than a third of those drugs were ultimately deprescribed after twelve months. These results highlight the difficulty of the deprescribing process even with well-accepted supporting tools.

In the case of Proton Pump Inhibitors (PPIs), we observed that the tools considered them PIMs in 40% of cases. After analysing the results, we observed that the tools only considered PPIs appropriate if they were prescribed in patients with a gastroduodenal ulcer or esophagic reflux without considering antiplatelet or anticoagulant treatments. There is a lack of consensus in actual guidelines regarding this matter, which is the reason why the tools have not adopted this item yet. However, studies have shown a decrease in gastrointestinal bleeding amongst the population over 65 years of age taking antiplatelets/anticoagulants [26]. Thus, we decided to unify these criteria by adding patients with concomitant treatment with antiplatelets or anticoagulants. Then, we observed that the proportion of PPIs considered potentially inappropriate was 16.9%.

After comparing the four tools, we observed a tendency to consider some TGs, such as hypnotic/anxiolytics, antidepressants, and antipsychotics, as PIMs; however, some differences between tools regarding other TGs were noticed. While BEERS and STOPP proposed a significant number of PPIs to be deprescribed (38% and 24%, respectively), PRISCUS and LESS-CHRON did not. The same situation was observed with urate-lowering drugs, for which LESS-CHRON proposed 59.4% of them to be deprescribed, while none of the other tools did. This brings to light that more unified criteria may be needed to support GPs in this task.

This study has some limitations. During the 12-month period, six out of fourteen patients’ quotas had at least one GP transfer. The deprescribing process usually requires a relationship of trust between the patient and the doctor. Furthermore, medication optimisation is a patient-centred process in which the prescriber and the patient interact. Thus, these two reasons might have caused a reduction in the outcomes of deprescribed drugs. The COVID-19 pandemic also might have had an impact on this matter. From March to May 2020, the organisation of the medical centre moved to telehealth (except for emergencies) as part of the contingency plan. Furthermore, individual agendas were suspended, and electronic prescriptions were allowed to be renewed for more extended periods than usual. The study was carried out in one PHC covering 27,000 inhabitants; the results cannot be extrapolated to the general population. Deprescribing tools are generalised criteria to identify PIMs that may not apply to all patients. Thus, medical judgement is essential regardless of the tool outcome. It is also possible that the natural tendency of polymedicated patients is to reduce medication, considering ten drugs as the cut-off point (effect of regression toward the mean).

This study also has some strengths. A comprehensive evaluation of the medical records of excessively Polymedicated patients was performed. Additionally, this deprescribing was studied in clinical practice, with the patient’s GP reviewing their medication; regardless of the tool outcome for one drug, medical judgment prevailed. Most studies on deprescribing measure a single drug or TG. A significant number of studies on several deprescribing TGs are performed in nursing homes, community dwellings, or hospitals during admission [17]. The Spanish elderly population tends to live in their own homes or with their families during the last decades of their lives, and their medication reviews are performed by their GPs. Since this is the most common scope in some countries, this study was designed to measure the number of PIMs among this population in this particular scope.

More than thirty PIM lists with explicit criteria have been published, gathering more than nine hundred medications, but only a few drugs are common to all lists [27]. Most studies on identifying PIMs are carried out in Europe; therefore, the most used deprescribing criteria are BEERS and STOPP. Very few of those studies have combined several deprescribing criteria [15]. As was observed in this study, only a few TGs were common to the four explicit criteria used. If PIM identification is based on a single criterion, some PIMs might be missed. We believe that unifying deprescribing criteria should be considered in the future to make the deprescribing process more efficient. Some online tools, such as CheckTheMeds, have started this task [19], and perhaps it would be convenient to include them in primary care computer programs as another supporting tool. Explicit deprescribing criteria are just a supporting tool used by doctors, who should have the final say in the deprescribing process. We did not find another study that measured the level of agreement of the GPs with the explicit criteria. Randomised control trials on deprescribing PIMs have shown that the withdrawal process is carried out less than expected. Deprescribing PIMs is a complex and time-consuming task. Explicit criteria and supporting tools should be easy to apply. Many explicit criteria have been published and proven effective, and some tools are trying to unify these criteria to facilitate this task, but the time used to apply them is still long. More research in this field is needed in order to find a more efficient way to review medication in clinical practice in the context of elevated healthcare pressure.

## 5. Conclusions

Elderly excessively Polymedicated patients accumulate a great number of PIMs. Despite the present recommendations for less intensive treatment in elderly patients, we observed a tendency to maintain treatments. The use of deprescribing supporting tools, such as explicit criteria, is feasible in primary care, and these tools are well-accepted by GPs. However, in less than half of the cases, eventual withdrawal was carried out. A great number of explicit deprescribing criteria have been published. However, the level of concordance between them is low, and using a single tool can lead to missing PIMs in some cases. Online tools that combine several explicit criteria could be an option to make the deprescribing process more efficient in the future.

## Figures and Tables

**Figure 1 ijerph-19-13241-f001:**
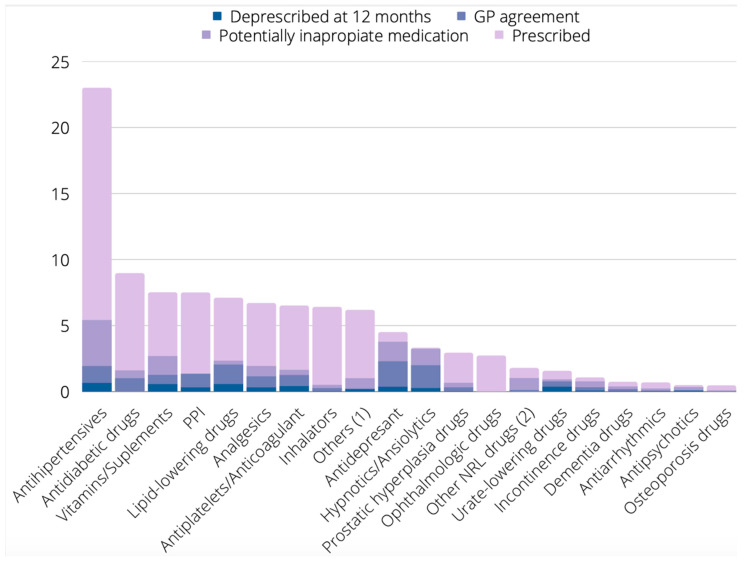
Percentage of prescribed medication, potentially inappropriate medication, level of agreement of the GP with the deprescribing proposal, and drugs eventually deprescribed at 12 months per therapeutic group. ^(1)^ Antihistamine drugs, drugs for vestibular disorders, drugs for cancer treatment, etc. ^(2)^ Other drugs for neurological disorders, such as epilepsy, Parkinson’s disease, etc.

**Table 1 ijerph-19-13241-t001:** Characteristics of excessively polymedicated patients over 75 years old.

Characteristic	Total, n/N (%)
Age ^1^	82 (78–85)
Female	97/168 (57.7)
Number of drugs ^2^	12.24 (±2.17)
PIMs per patient ^3^	3.86 (±2.56)
Comorbidities ^4^	8 (6–10)
Hypertension	157/168 (91.8)
Dyslipidemia	123/168 (73.2)
Diabetes	93/168 (54.4)
Depression/Anxiety	82/168 (48)
Chronic musculoskeletal disease	68/168 (40.5)
Urinary incontinence	61/168 (36.3)
Respiratory disease	60/168 (35.7)
Arrhythmias	56/168 (32.7)
Chronic kidney failure	54/168 (32.1)
Cardiac insufficiency	54/168 (32.1)
Ischemic cardiomyopathy	49/168 (28.7)
Prostatic hyperplasia	48/71 (67.6) ^a^
Insomnia	43/168 (25.1)
Hyperuricemia	40/168 (23.4)
Chronic dermatologic disease	30/168 (17.9)
Chronic ophthalmologic disease	30/168 (17.9)
Heart valvular disease	28/168 (16.4)
Peripheral artery disease	26/168 (15.5)
Thyroid disease	25/168 (14.9)
Cerebrovascular disease	24/168 (14.3)
Gastroesophageal reflux disease	23/168 (13.7)
Rheumatologic disease	21/168 (12.3)
Dementia	17/169 (10.1)
Gastroduodenal ulcer	10/168 (6)
Liver disease	5/168 (3)
Other ^5^	51/168 (30.4)

^1,4^ Median (IQR, interquartile range). ^2,3^ Standard deviation (95% CI). ^5^ Neoplasia, epilepsy, otologic and vestibular disease, bile duct and pancreatic pathology, hematologic disease, inflammatory bowel disease. ^a^ N = male patients.

**Table 2 ijerph-19-13241-t002:** Level of control of excessively Polymedicated patients in some of the pathologies.

Clinical Feature (Unit)	Associated Diagnosis	N	Average ^1,2^	Patients in Range		Nº of Drugs Per Diagnosis N (%)		
				Parameter ^1^	N (%)	1	2	3
BP (mmHg)	Hypertension	157	132/72	BP < 140/90	88 (56.1)	14 (8.9)	42 (26.8)	100 (63.7)
				BP < 150/90	117 (74.5)			
HbA1C (%)	Diabetes	93	7	HbA1C < 8	69 (75)	33 (35.5)	27 (29)	27 (29)
Total cholesterol (mg/dL)	Dyslipidemia	123	175.5	Total cholesterol < 200 and/or LDLc < 130	82 (77.4)	108 (87.8)	15 (12.2)	0 (0)
LDL cholesterol (mg/dL)			96.7					
Urate (mg/dL)	Hyperuricemia	40	6.2	Urate ≤ 6	15 (37.5)			

BMI (Kg/m^2^)		166	29.7	BMI < 30	89 (53)			
GF (L/min)		166	56.1	GF ≥ 60	76 (45.2)			

BMI = Body mass index, GF = Glomerular filtrate. ^1^ The units in this column are expressed in the first column. ^2^ The average of each clinical feature was calculated amongst the patients with the diagnosis described in the second column, except for the BMI and glomerular filtrate, which were calculated among the sample.

**Table 3 ijerph-19-13241-t003:** Prescribed drugs per therapeutic group and drugs proposed for deprescribing per therapeutic group.

Therapeutic Group	Prescribed N (%)	Proposed for Deprescribing by Any Tool N (%)	Proposed for Deprescribing by BEERS N (%)	Proposed for Deprescribing by LESS-CHRON N (%)	Proposed for Deprescribing by STOPP N (%)	Proposed for Deprescribing by PRISCUS N (%)
Antihypertensives	468 (22.9)	111 (23.7%)	44 (9.4)	58 (12.4)	35 (7.5)	1 (0.2)
Antidiabetic drugs	182 (8.9)	33 (18.1)	6 (3.3)	27 (14.8)	6 (3.3)	1 (0.5)
Vitamins/Supplements	153 (7.5)	55 (35.9)	0 (0)	16 (10.5)	34 (22.2)	0 (0)
PPI	150 (7.3)	25 (16.9)	57 (38)	1 (0.7)	36 (24)	0 (0)
Lipid-lowering drugs	145 (7.1)	48 (33.1)	9 (6.2)	47 (32.4)	3 (2.1)	0 (0)
Analgesics	137 (6.7)	40 (29.2)	36 (26.3)	1 (0.7	24 (17.5)	0 (0)
Antiplatelets/Anticoagulants	133 (6.5)	34 (25.6)	15 (11.3)	28 (21.1)	5 (3.8)	0 (0)
Inhalators	131 (6.4)	11 (8.4)	12 (9.2)	0 (0)	10 (7.6)	0 (0)
Others ^1^	126 (6.2)	21 (16.7)	3 (3.2)	0 (0)	10 (7.9)	1 (0.8)
Antidepressants	92 (4.5)	77 (83.7)	49 (53.3)	70 (76.1)	11 (12)	3 (3.3)
Hypnotics/Anxiolytics	68 (3.3)	66 (97.1)	64 (94.1)	60 (88.2)	59 (86.8)	37 (57.8)
Prostatic hyperplasia drugs	60 (2.9)	14 (23.3)	5 (8.3)	1 (1.7)	11 (18.3)	1 (1.7)
Ophthalmologic drugs	55 (2.7)	1 (1.8)	0 (0)	0 (0)	1 (1.8)	0 (0)
Other NRL drugs ^2^	36 (1.8)	21 (58.3)	12 (33.3)	0 (0)	2 (5.6)	0 (0)
Urate-lowering drugs	32 (1.6)	19 (59.4)	0 (0)	19 (59.4)	2 (6.3)	0 (0)
Incontinence drugs	22 (1.1)	16 (72.7)	6 (27.3)	2 (9.1)	9 (40.9)	4 (18.2)
Dementia drugs	15 (0.7)	8 (53.3)	1 (6.7)	6 (40)	3 (20)	0 (0)
Antiarrhythmics	14 (0.7)	5 (35.7)	5 (35.7)	1 (7.1)	3 (21.4)	3 (21.4)
Antipsychotics	10 (0.5)	7 (70)	5 (50)	2 (40)	5 (50)	1 (20)
Osteoporosis drugs	9 (0.4)	2 (22.2)	0 (0)	0 (0)	2 (22.2)	0 (0)
**All groups**	**2038 (100)**	**649 (31.8)**	**329 (16.1)**	**339 (16.6)**	**271 (13.3)**	**52 (2.6)**

^1^ Antihistamine drugs, drugs for vestibular disorders, drugs for cancer treatment, etc. ^2^ Other drugs for neurological disorders, such as epilepsy, Parkinson’s Disease, etc.

## Data Availability

The data used in this study cannot be publicly shared because of restrictions imposed by the institution (Son Serra-La Vileta Primary Healthcare Center) and asserted by the Primary Care Research Committee and the Balearic Ethical Committee.

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
