# Peer review of "LESS-PHARMA Study: Identifying and Deprescribing Potentially Inappropriate Medication in the Elderly Population with Excessive Polypharmacy in Primary Care"

_ijerph, 2022, doi:10.3390/ijerph192013241_

Round 1

Reviewer 1 Report

Dear authors:

After Reading the manuscript entitle “LESS-PHARMA study. Identifying and deprescribing potentially inappropriate medication in the elderly population with excessive polypharmacy in primary Care”, I consider that the topic of the paper could be interesting and relevant. Inappropriate prescribing in the elderly is a risk factor for higher adverse drugs reactions, hospitalization and mortality rates. Therefore, I agree with the authors on the need to identify irrational prescriptions and implement interventions to improve geriatric clinical practices in Primary Healthcare Centre.

I consider the Discussion adequate and I appreciate the limitations pointed out by the authors in their review article. I agree with them, but also with the strengths.

Major comments:

I recommend authors to follow the current rule regarding the use of decimal separators. The decimal separator must be changed from " , " to " . " in all tables, as well as throughout the text of the manuscript.

In order to improve the understanding of the article, I recommend the authors include a subsection “Statistical Analysis” on the section Materials and Methods.

On the other hand, and as a personal recommendation with the aim of turning this review article into a good review, I encourage authors to modify table 3: perhaps it would be interesting to place the Therapeutic groups in alphabetical order or better still by percentage of prescription, from highest to lowest.

Line from 200 to 204: It would be interesting for understanding to indicate the cases proposed in each case.

Figure 1 is not referenced throughout the text. Please correct this error

I also recommend the authors, modify the graph in Figure 1 for for better interpretation and visualization. Value a change of scale or graph.

Minor comments:

Line 17: Enter the abbreviation of Potentially inappropriate medication (PIMs), since line 29 refers to it.

Line 37 and 38: These statements may need to be expanded or write differently.

Line 78: Perhaps it would be interesting to include in the term STOPP-START the following clarification: Screening Tool of Older people´s prescriptions-Screening Tool to Alert to Right Treatment.

Line 104: I invite the authors to consider the option of indicating the number of patients selected.

Line 117: Specify version, commercial company and place for CheckTheMeds.

Line 148: Perhaps it would be interesting to include the clarification: … “were included in our study after applying the inclusion and exclusion criteria”.

Line 157: I would put a period and followed after (97.6%).

Line 186: Please unify to one or two decimals.

Other comments:

I would be grateful to the authors of the manuscript if they could clarify the term of the line 98: Prescriptions “on demand”.

Author Response

Reviewer 1

I consider the Discussion adequate and I appreciate the limitations pointed out by the authors in their review article. I agree with them, but also with the strengths.

Dear Reviewer,

Thank you very much for reviewing the manuscript, for your positive feedback, and for providing us with constructive comments that improved the quality of the paper. We agree with all of your comments and thus, we now submit a revised version of the manuscript.

Major comments:

I recommend authors to follow the current rule regarding the use of decimal separators. The decimal separator must be changed from " , " to " . " in all tables, as well as throughout the text of the manuscript.

Thank you for your comment. All the decimals separators have been changed in the new version.

In order to improve the understanding of the article, I recommend the authors include a subsection “Statistical Analysis” on the section Materials and Methods.

Thank you for your comment. Subsections have been now added to the section Material and Methods.

On the other hand, and as a personal recommendation with the aim of turning this review article into a good review, I encourage authors to modify table 3: perhaps it would be interesting to place the Therapeutic groups in alphabetical order or better still by percentage of prescription, from highest to lowest.

Thank you so much for your recommendation to help to elevate this article. Therapeutic groups have been placed by percentage of prescription in table 3.

Line from 200 to 204: It would be interesting for understanding to indicate the cases proposed in each case.

Thank you so much for your comment. We agree that it would be better to put the number of drugs agreed to deprescribe in this case. We have added this information.

Figure 1 is not referenced throughout the text. Please correct this error

Thank you so much for detecting this. It has been amended in the new version.

I also recommend the authors, modify the graph in Figure 1 for for better interpretation and visualization. Value a change of scale or graph.

Thank you for your recommendation. We have changed the type of graph and we believe that it better summarizes the information.

Minor comments:

Line 17: Enter the abbreviation of Potentially inappropriate medication (PIMs), since line 29 refers to it.

Thank you for your suggestion. We changed it.

Line 37 and 38: These statements may need to be expanded or write differently.

Thank you for your suggestion. We changed it.

Line 78: Perhaps it would be interesting to include in the term STOPP-START the following clarification: Screening Tool of Older people´s prescriptions-Screening Tool to Alert to Right Treatment.

Thank you for your suggestion. We added it.

Line 104: I invite the authors to consider the option of indicating the number of patients selected.

Thank you for you suggestion. This information is specified in subsection Participants in the Results section.

Line 117: Specify version, commercial company and place for CheckTheMeds.

Thank you for your suggestion. We have added this information in the section Material and Methods.

Line 148: Perhaps it would be interesting to include the clarification: … “were included in our study after applying the inclusion and exclusion criteria”.

Thank you for your recommendation. We changed it.

Line 157: I would put a period and followed after (97.6%).

Thank you for your recommendation. We changed it.

Line 186: Please unify to one or two decimals.

 Thank you for your suggestion. We have unified it to two decimals.

Other comments:

I would be grateful to the authors of the manuscript if they could clarify the term of the line 98: Prescriptions “on demand”.

Thank you for your suggestion. “On demand” medication is prn medication or medication taken only when needed. We have added this information.

Thank you very much for reviewing our manuscript!

Reviewer 2 Report

To the authors:

The article is easy to read and the objectives are very clear, In the part of the identification of potentially inappropriate medication it makes a very good contribution of the use of different instruments and their comparison. It also presents a computer application for use in clinical practice. There is a good description of the degree of control of cardiovascular risk factors.

The deprescription part has some weaknesses that the authors should comment on in the paper.

Introduction

Line 74:

When discussing medication review, a few comments should be added.

Medication review, in its broad sense, takes into account the drug regimen, clinical assessment criteria and the patient's goals and priorities.

It should also take into account that it is a complex process, as multiple factors interact to optimise medication, both at the level of the healthcare system, professionals and patients.

In this section, it should be noted that specific competencies are needed to carry out this medication review process.

Medication review is a high-value practice that should be carried out in any care setting that has the competencies to do so. In geriatric services, medication review is a common practice.

Material and Methods

line 125

It would be important to reflect on what criteria it was decided whether the primary care physician agreed or disagreed with the proposed changes. It would be important to know whether he/she was trained in geriatric pharmacology, whether other aspects such as the patient's frailty were taken into account or whether there was a discussion with the patient.

Results

line 147

Community studies provide a lot of epidemiological information, in which case, with unrestrictive exclusion criteria, it is notable that 120 patients did not enter the study. It would be necessary to have information on this group if it represents a selection bias. In the end, we are intervening in 0.6% of the population.

In the 6-month follow-up only one loss due to mortality was detected.

line 195 .

It raises the doubt that one of the main criteria for non-agreement in the de-prescription process is the lack of control of the pathologies, when in the clinical description there is very good control.

The methodology describes that the anticholinergic burden is available and no results are given.

Discussion

Line 242.

The instruments used in this article are very good at identifying potentially inappropriate prescribing, but do not provide any information on how to apply these data to each individual patient.

Line 262.

In the section on limitations, it should be noted that the medication optimisation process has not considered the barriers to its implementation.

Medication optimisation is a patient-centred process in which prescriber and patient aspects interact.

All this justifies that the number of medication withdrawals is very low.

It would be good to compare with other studies

Author Response

Reviewer 2

The article is easy to read and the objectives are very clear, In the part of the identification of potentially inappropriate medication it makes a very good contribution of the use of different instruments and their comparison. It also presents a computer application for use in clinical practice. There is a good description of the degree of control of cardiovascular risk factors.

Dear Reviewer,

Thank you very much for reviewing the manuscript, for your positive feedback, and for providing us with constructive comments that improved the quality of the paper. We agree with all of your comments and thus, we now submit a revised version of the manuscript.

The deprescription part has some weaknesses that the authors should comment on in the paper.

Introduction

Line 74:

When discussing medication review, a few comments should be added.

Medication review, in its broad sense, takes into account the drug regimen, clinical assessment criteria and the patient's goals and priorities.

It should also take into account that it is a complex process, as multiple factors interact to optimise medication, both at the level of the healthcare system, professionals and patients.

In this section, it should be noted that specific competencies are needed to carry out this medication review process.

Medication review is a high-value practice that should be carried out in any care setting that has the competencies to do so. In geriatric services, medication review is a common practice.

Thank you so much for your suggestions. These comments have been added to the new version of the manuscript.

Material and Methods

line 125

It would be important to reflect on what criteria it was decided whether the primary care physician agreed or disagreed with the proposed changes. It would be important to know whether he/she was trained in geriatric pharmacology, whether other aspects such as the patient's frailty were taken into account or whether there was a discussion with the patient.

Thank you so much for your suggestion. The criteria that every GP used to decide if they agree with the proposal was based on the medical history, blood test results and the social and emotional background of each patient. In Spain, GPs treat their patients throughout their lives and acquire a very detailed knowledge of the clinical, social and emotional background. A patient-centered and integrated care model is essential in the deprescribing process and a comprehensive of the clinical profile, functional status, mental health and socio-economic status, considering also patient preferences, is key to ensuring the optimal clinical management of our patients. We have clarified this in the new version of the manuscript.

Pharmacological management and rational use of medications are part of the training during the registrar period of the GPs and it is a competence that continues developing during the daily professional activity of each GP.

Results

line 147

Community studies provide a lot of epidemiological information, in which case, with unrestrictive exclusion criteria, it is notable that 120 patients did not enter the study. It would be necessary to have information on this group if it represents a selection bias. In the end, we are intervening in 0.6% of the population.

Thank you a lot for your comment. The 120 patient that did not enter the study met the exclusion criteria of being followed by the private sector. Medications of these patients are mainly reviewed by other doctors in the private sector and GP has a little roll modifying these treatments.

In the 6-month follow-up only one loss due to mortality was detected.

That is correct

line 195 .

It raises the doubt that one of the main criteria for non-agreement in the de-prescription process is the lack of control of the pathologies, when in the clinical description there is very good control.

Thank you so much for your observation. It is true that our patient sample have very good level of control of cardiovascular risk factors, but that only includes 3 of the 21 therapeutic groups analysed (antidiabetic, lipid-lowering drugs and antihypertensive drugs). Also, antihypertensive drugs are common treatment of heart failure.

The methodology describes that the anticholinergic burden is available and no results are given.

Thank you for your comment. Checkthemeds proposed drugs with anticholinergic burden for deprescribing when this one was elevated in a patient, but the level of anticholinergic burden has not been recorded as a variable in this study.

Discussion

Line 242.

The instruments used in this article are very good at identifying potentially inappropriate prescribing, but do not provide any information on how to apply these data to each individual patient.

Thank you very much for your comment. We agree that this tools are very useful at identifying potentially inappropriate medications but they have to be applied and interpreted by a well-trained health care professional following a patient-centered and integrated care model.

Line 262.

In the section on limitations, it should be noted that the medication optimisation process has not considered the barriers to its implementation.

Medication optimisation is a patient-centred process in which prescriber and patient aspects interact.

All this justifies that the number of medication withdrawals is very low.

Thank you for your suggestion. We have added this as a limitation in the new version

Thank you very much for reviewing our manuscript!
